# Electrographic Flow Mapping Provides Prognosis for AF Ablation Outcomes Across Two Independent Prospective Patient Cohorts

**DOI:** 10.3390/jcm14030693

**Published:** 2025-01-22

**Authors:** Kent R. Nilsson, Amitesh Anerao, Melissa H. Kong, Pawel Derejko, Tamás Szili-Török, Sandeep Goyal, Mohit Turagam, Atul Verma, Steven Castellano

**Affiliations:** 1Department of Cardiac Electrophysiology, Piedmont Heart Institute, Athens, GA 30309, USA; nilsson@uga.edu (K.R.N.); sandeep.goyal@piedmont.org (S.G.); 2Medical College of Georgia, Augusta University/University of Georgia Partnership, Athens, GA 30602, USA; amitesh.anerao@uga.edu; 3Cortex, Inc., Menlo Park, CA 94025, USA; mhkong1@gmail.com; 4Department of Cardiology, Medicover Hospital Warsaw, 02-972 Warsaw, Poland; pawel.derejko@medicover.pl; 5Cardiology Center, Department of Internal Medicine, University of Szeged, 6720 Szeged, Hungary; szilitorok@hotmail.com; 6Icahn School of Medicine at Mount Sinai, New York, NY 10029, USA; mohit.turagam@mountsinai.org; 7McGill University Health Centre, Montreal, QC H4A 3J1, Canada; atul.verma@mcgill.ca

**Keywords:** ablation, arrythmias, atrial fibrillation, basket catheter, cardiac, electrographic flow mapping, panoramic mapping

## Abstract

**Background/Objectives:** Electrographic flow (EGF) mapping allows for the visualization and quantification of atrial fibrillation (AF) wavefront propagation patterns. EGF-identified sources were shown in the randomized controlled *FLOW-AF* trial to significantly increase the likelihood of AF recurrence within 1 year if left unablated. Electrographic flow consistency (EGFC) additionally measures the stability of observed wavefront patterns, such that patients with more organization have a healthier substrate and lower recurrence. Source presence and EGFC can be used collectively to assign mechanistic phenotypes to AF patients. **Methods:** The patient phenotypes, treatment modalities, and outcomes in *FLOW-AF* were compared with those of patients in the ensuing *AF-FLOW Global Registry*, which was conducted by separate physicians at discrete clinical centers. **Results:** Patients with low EGFC (≤0.62) had a 12-month freedom from AF (FFAF) of 46%, while those with a high mean EGFC (>0.62) had a FFAF of 81%. Right atrial EGFC was correlated with left atrial EGFC, and the highest recurrence occurred in those with biatrial low EGFC. Source presence also affected the recurrence rates in both trials, such that the presence of EGF-identified sources in PVI-only patients lowered the FFAF from 65% to 36%, but the elimination of sources produced a 30% absolute increase in FFAF from 36% to 66%. **Conclusions:** Patient outcomes by EGF-based AF phenotype were consistent across two cohorts of patients from separate clinical trials at distinct centers. Patients with a high EGFC and no sources post-procedure had the best outcomes. EGF mapping provides insights into underlying disease pathophysiology and may be employed prospectively to predict recurrence.

## 1. Introduction

Atrial fibrillation (AF) is the most common arrhythmia and is associated with significant morbidity and mortality and a rising prevalence worldwide [1,2,3,4]. AF patients have a high risk for thromboembolism, stroke, myocardial infarction, hospitalization, and death, which is not significantly reduced even among asymptomatic patients [5]. This is because asymptomatic patients—who account for up to 40% of the diagnosed population of AF patients [6]—more often progress to permanent AF in the absence of appropriate anti-arrhythmic drugs or ablation therapy, which often curtail complications [5].

While the need for more widespread AF screening and treatment is growing, there remains a ceiling of efficacy observed in the effectiveness of AF therapies. Pulmonary vein isolation (PVI) has, thus far, emerged as the cornerstone of management for both paroxysmal AF (PAF) and persistent AF (PeAF). Though recurrence rates following PVI are significantly higher for PeAF than for PAF, empiric standardized endocardial ablation lesion sets for PeAF have failed to improve clinical outcomes beyond PVI alone [7,8]. As a result, PVI without adjunctive ablation has remained the gold standard for the treatment of PeAF.

Given the challenges of managing patients with PeAF, multiple mapping and ablation strategies have been proposed to improve patient outcomes. Some have focused on improving the effectiveness of PVI ablations. For example, the recent CHARISMA trial showed that successful PVI could be predicted when local impedance dropped by at least 21 Ω at anterior sites and 18 Ω at posterior sites, and that higher contact forces were non-linearly correlated with increased impedance drops [9]. Such findings add to the prior understanding of the importance of good contact in preventing recurrence from the TOCCATA study [10]. Additionally, pulsed field ablation (PFA) has been vetted as a novel, tissue-selective alternative to radiofrequency (RF) ablation with a selectivity for cardiac tissue that reduces the risk for esophageal injuries, pulmonary vein (PV) stenosis, and phrenic nerve palsy [11,12,13,14,15,16,17]. Moreover, PFA trials have, thus far, shown a high 76% durability of PVIs among recurring patients, as well as similar efficacy, with 60% ± 10% freedom from arrythmias after one year for PeAF patients [11,18,19].

Other strategies for improving AF ablation outcomes have, instead, emphasized additional lesion sets that could improve outcomes beyond PVI. These include complex fractionated atrial electrogram ablation [20], rotor ablation [21], charge density mapping [22], dispersion mapping [23], image-guided fibrosis [24], and artificial-intelligence-driven lesions [22]. However, each method has substantial limitations and variable clinical results.

To this end, electrographic flow (EGF) mapping has been shown to be a promising modality that rapidly (1) localizes extra-PV drivers of persistent AF and (2) quantifies the health of the underlying substrate as electrographic flow consistency (EGFC). Most mapping technologies suffer from a poor spatial resolution, difficulty in identifying the true source of the arrhythmia due to complex electrical patterns, susceptibility to noise and artifacts, a lack of real-time mapping capabilities, operator dependence in interpreting data, and inconsistent results across different mapping techniques or a lack of reproducibility on repeat mappings [25,26,27]. However, EGF mapping creates a full, near-real-time spatiotemporal reconstruction of organized atrial electrical wavefront propagation in order to reveal reproducible patterns pertaining to a patient’s rhythm-dependent map properties [28,29,30,31].

While avoiding many of the weaknesses of conventional mapping systems, EGF mapping has also led to improved procedural efficacy. The recent multicenter, randomized, controlled *FLOW-AF* clinical trial of 85 PeAF and long-standing PeAF (LS-PeAF) patients showed that EGF mapping identified sources in 60% of patients. The ablation of these sources in addition to PVI improved freedom from AF (FFAF) (68% vs. 17%, *p* = 0.042) and freedom from AF/Atrial Tachycardia (AT)/Atrial Flutter (AFL) (51% vs. 14%, *p* = 0.103) at 12 months [32]. Furthermore, source presence and EGFC could be used to phenotypically classify patients into four categories that had treatment-dependent outcomes post-ablation. The phenotypes were as follows: Type I—no extra-PV sources and a high EGFC; Type II—sources and a high EGFC; Type III—sources and a low EGFC; and Type IV—no sources and a low EGFC [32,33,34].

Subsequently, the *AF-FLOW Global Registry* study was performed to gather information on how physicians would employ EGF mapping in their clinical procedures and again assess post-procedure outcomes. The study consisted of 25 patients from five clinical centers with PAF, PeAF, and LS-PeAF. It again demonstrated that patients with a high EGFC (Types I and II) had higher rates of FFAF than those with a low EGFC (Types III and IV), and that patients with sources remaining post-procedure had a lower FFAF than those with no sources [35].

Here, we seek to compare outcomes across both *FLOW-AF* and the *AF-FLOW Global Registry*. We aim to use both studies to collectively analyze the predictive power of EGF mapping properties for post-procedure FFAF with a focus on (1) the impact of EGFC—both overall and by atrium—on 12-month outcomes, (2) the association between phenotype and outcome among patients receiving PVI-only procedures, and (3) the effect of source ablation on modulating these PVI-only outcomes.

## 2. Materials and Methods

### 2.1. Trial Overview

The details of the trials have been previously outlined for *FLOW-AF* (ClinicalTrials.gov identifier number NCT04473963) and the *AF-FLOW Global Registry* (NCT05481359). *FLOW-AF* enrolled 85 patients and was powered to test safety and map reproducibility for EGF-mapped patients [32]. The *AF-FLOW Global Registry* enrolled 25 patients and was powered to test source detection and ablation rate in an open clinical setting [35].

### 2.2. PVI Ablation Procedure

All procedures were performed under sedation after obtaining informed consent. The patients underwent 3-dimensional electroanatomic mapping with either Ensite NavX (Abbott, Abbott Park, IL, USA) or CARTO (Biosense Webster, Diamond Bar, CA, USA), followed by PVI using an irrigated-tip RF ablation catheter. The ablation catheter power and temperature settings were at the operator’s discretion. Intravenous heparin was administered for systemic anticoagulation to maintain an activated clotting time of >300 s prior to the insertion of the 64-pole basket mapping catheter. For any ablation of PVs, confirmation of PVI (entrance or exit block) was performed after a mandatory minimum 20-min wait following the last RF application.

### 2.3. EGF Mapping

EGF mapping was performed both pre- and post-PVI using the Ablamap^®^ v9.0.4-v11.1.0.56 software (Ablacon, Wheat Ridge, CO, USA). Unipolar electrograms (EGMs) were recorded using a commercially available 64-electrode basket mapping catheter connected to a proprietary, CE-marked recording system (EP Map, Herdecke, Germany). Each recording was 1 min long, and the unipolar signals were processed using a proprietary algorithm to remove the QRS complex, noise, baseline fluctuations, and far-field signals. Using a biharmonic spline interpolation based on Green’s function, the electrical field was estimated, and a Horn–Schunck flow estimation of the Green’s interpolation frames was performed to visualize atrial wavefront propagation over time. Haines et al. published an in-depth explanation of the theoretical basis for EGF mapping and details of the EGF mapping algorithm [36], which were more recently summarized by Castellano and Kong [34]. The dominant patterns of wavefront propagation over time were displayed as flow vectors in EGF Segment Maps, enabling the visualization of the origins of excitation, representing extra-PV AF triggers, called sources. Active sources exhibited divergent flow fields.

EGF mapping was typically performed by obtaining 1-min recordings from the 64-pole basket catheter placed in multiple standardized positions within each atrium, as well as additional recordings after any source ablations to confirm elimination. In the left atrium (LA), there were the following two standardized basket positions: (1) posterosuperior LA with the dome of the basket aiming toward the left PVs and LA appendage (LAA) and (2) lateral LA with the basket dome pointing toward the LAA and mitral valve. In the right atrium (RA), there were the following three standardized basket positions: (1) superior vena cava (SVC)/RA junction with the basket position halfway into the SVC; (2) central RA with the basket dome pointing to the SVC with a vertical catheter shaft orientation and the basket expanded to fill the RA; and (3) anterolateral RA with the basket dome pointing to the RA appendage and tricuspid valve and the catheter deflected anteriorly and laterally from the SVC. Mapping in additional basket positions in either atrium was performed if necessary to obtain complete endocardial atrial coverage.

### 2.4. EGF-Guided Ablation

If EGF mapping identified an extra-PV source with a source activity (SAC) of ≥ 26.5% based on the prespecified threshold for clinical significance, EGF-guided ablation at the target location of the focal source was required for patients in the treatment groups, though not for those in the control groups. Repeat EGF mapping was recommended to confirm the elimination of the source via reduction in the SAC below the SAC threshold. If additional sources were unmasked during repeat mapping, additional ablation was advised until all sources were eliminated.

### 2.5. EGFC Calculations from EGF Maps

EGFC was computed for each recording from the Euclidean length of vector field estimates across all basket electrodes over time, thereby providing a measure of the overall magnitude of the flow across the mapping catheter, averaged in arbitrary units. Importantly, electrodes with low contact were removed from the calculation because of artificially low local EGFC values that were observed at non-contacting electrode positions.

After calculating the EGFC for each recording, an overall EGFC was then computed for each patient in AF and/or SR. To perform this, the EGFCs from each of a patient’s recordings in a given basket position during the chosen rhythm were first averaged to compute positional EGFC means per rhythm per patient. These positional EGFCs were then averaged within each atrium to generate a mean EGFC per atrium per patient. Such a method allowed for all recording positions to be given equal weight. Lastly, each patient’s two atrial EGFCs were then averaged again to generate an overall mean EGFC per patient. A threshold of 0.62 for the overall mean EGFC was used for phenotyping high vs. low EGFC, as derived from prior studies [32,33].

### 2.6. Statistical Analysis

Summary data are expressed as mean value ± one standard deviation. *p*-values for group comparisons were computed using two-tailed, independent *t*-tests for continuous variables and two-tailed z-tests for proportions, as appropriate. For all statistical tests, the null hypothesis was rejected at the level of *p* < 0.05 (bolded).

## 3. Results

### 3.1. Trial Design and Demographics

The two trials that were analyzed were *FLOW-AF* and the *AF-FLOW Global Registry*. *FLOW-AF* was a multi-centered, randomized trial evaluating EGF mapping prospectively for the first time. Redo and PeAF or LS-PeAF patients were analyzed from Homolka Hospital (Prague, Czech Republic), Praxisklinik Herz und Gefaesse (Dresden, Germany), the Erasmus MC Department of Cardiology (Rotterdam, The Netherlands), and the University Medical Center Hamburg-Eppendorf (Hamburg, Germany). Patients with sources with an SAC of ≥ 26.5% were randomized to either treatment, where the sources were ablated, or control where the sources were left unablated.

Subsequently, the *AF-FLOW Global Registry* was conducted as a follow-up study to explore the implications of EGF mapping in a broader patient population. The *AF-FLOW Global Registry* consisted of 25 prospective all-comers, including de novo and PAF patients. Patients were enrolled from University Health Center, the University of Georgia (Athens, Georgia), Piedmont Athens Regional Hospital (Athens, Georgia), Mount Sinai Hospital (New York, NY, USA), the Erasmus MC Department of Cardiology (Rotterdam, The Netherlands), and Medicover Hospital (Warsaw, Poland). The *AF-FLOW Global Registry* also gave physicians the liberty to treat EGF-identified sources in a way they saw fit, and patients were not randomized to have all sources with an SAC of ≥ 26.5% ablated or left unablated as they were in the *FLOW-AF* study arms.

Patient demographics by trial are shown in Table 1. There were no significant differences among any of the demographic variables between the two trials.

### 3.2. EGFC Was Related to AF Recurrence Rates Across All Patient Cohorts

Substrate health was first studied across the FFAF vs. recurrence patients in the two studies. When analyzing the EGFCs across all patients, those who experienced FFAF had mean EGFC values of 0.52 ± 0.11 and 0.59 ± 0.15 in *FLOW-AF* and the *AF-FLOW Global Registry*, respectively. By contrast, patients who experienced AF recurrence had mean EGFC values of 0.49 ± 0.09 and 0.47 ± 0.12. The pooled total EGFCs across the two studies demonstrated that patients who experienced FFAF had a mean EGFC of 0.54 ± 0.12, while recurring patients had a mean EGFC of 0.49 ± 0.09 (***p* = 0.022**). EGFC differences by outcome are shown in Table 2.

Moreover, the recurrence rates were similar across the two trials based on the EGFC classification of all-comers. Patients with a low overall EGFC (≤0.62) demonstrated FFAF rates of 42% (n = 48) and 59% (n = 17) in *FLOW-AF* and the *AF-FLOW Global Registry*, respectively, while those with a high mean EGFC (>0.62) had FFAF rates of 79% (n = 28) and 100% (n = 4). Pooled data across both trials showed that those with a low overall EGFC had a total FFAF rate of 46% (n = 65) vs. 81% (n = 32) for those with a high overall EGFC (***p* = 0.005**), as shown in Figure 1.

Further analysis was conducted on EGFC by atrium among patients with sources ablated by the end of the procedure. Patients with a high atrial EGFC were defined as having a mean RA EGFC ≥ the population mean RA EGFC or a mean LA EGFC ≥ the population mean LA EGFC for each atrium, respectively. Those with a high EGFC in both the right and left atria showed the highest FFAF of 87% (n = 15). For patients who presented with a high mean EGFC in just the right or left atrium, their FFAF rates were 67% (n = 9) and 63% (n = 8), respectively. Lastly, patients who had a low EGFC in both right and left atria had an FFAF rate of 55% (n = 31). Therefore, patients with a high mean EGFC in both atria showed a 32% increase in FFAF compared with those with a low mean EGFC in both atria (***p* = 0.034**). Recurrence by mean EGFC in each atrium is shown in Figure 2 alongside dashed lines indicating the population mean EGFC for each atrium.

Across all patients, the mean EGFC was lower in the LA than the RA (0.53 ± 0.19 vs. 0.60 ± 0.23, ***p* = 0.026**). A linear relationship was also established between the mean RA EGFC and mean LA EGFC of patients (m = 0.51, r = 0.42, ***p* < 0.001**), which was unaffected by trial or recurrence status.

### 3.3. Phenotyping by EGFC and Source Presence Can Aid in Predicting Procedure Outcomes

The four phenotypes and the characteristic appearances of their EGF Summary Maps and EGF Segment Maps are shown in Figure 3, and overall outcomes by phenotype are shown in Figure 4. Table 3 follows with outcomes by phenotype and treatment modality.

Trends in the 12-month FFAF rates among the four phenotypes remained consistent across both trials. Among PVI-only patients noted in Table 3, Type I patients with no EGF-identified sources and a healthy substrate had 100% FFAF rates in both *FLOW-AF* and the *AF-FLOW Global Registry*. Type II patients with sources but a healthy substrate had a worse FFAF rate of 50% in *FLOW-AF*, though no Type II patients received PVI-only in the *AF-FLOW Global Registry*. This outcome was similar to that of Type IV patients with no sources but an unhealthy substrate, who had FFAF rates of 50% and 57%, respectively. Lastly, Type III patients with EGF-identified sources and an abnormal substrate had the worst outcomes in both trials, with FFAF rates of 23% and 50%. The pooled data of the two trials showed Type I patients to have 100% FFAF, Type IV having 52% FFAF, Type II having 50% FFAF, and Type III having 27% FFAF. Therefore, as PVI-only patients demonstrated either sources or a low EGFC in moving from Type I with neither to Types II or IV with one such factor, the FFAF significantly decreased (***p* = 0.003**). It decreased further in Type III patients with both factors compared to Types II or IV with just one, such that the overall differences in FFAF were most pronounced in Type I vs. Type III (***p* < 0.001**).

### 3.4. Source Ablation Improves FFAF in Both Independent Studies

When the outcomes by phenotype across the two studies shown in Figure 4 were analyzed, the *AF-FLOW Global Registry* FFAF rates were similar to the *FLOW-AF* FFAF rates among Type I and Type IV patients without sources, but the FFAF was noticeably higher among *AF-FLOW Global Registry* Type II and Type III patients with sources. Moreover, within the *AF-FLOW Global Registry*, Type I outcomes were identical to Type II outcomes at 100% FFAF, while Type III and Type IV outcomes were also highly similar at 60% and 57% FFAF, respectively. These diverging results between the trials were believed to predominantly result from procedural differences. In contrast with the *FLOW-AF* protocol requiring the randomization of source patients to PVI-only or PVI + EGF-guided ablation of sources, the *AF-FLOW Global Registry* patients predominantly received PVI plus targeted adjunctive source ablation when sources were observed. 

Data were consequently also broken down by the combination of source presence and treatment modality. Among patients receiving PVI-only, source presence lowered the FFAF from 65% in pooled Type I and Type IV patients to 36% in pooled Type II and Type III patients (***p* = 0.023**). Moreover, among the source-containing Type II and Type III patients, those who received EGF-guided source ablation in addition to PVI demonstrated significantly improved outcomes, with 91% (10/11) Type II patients and 52% (11/21) Type III patients having FFAF, up from 50% (5/10) and 27% (4/15) after PVI alone. Combined data across both studies accordingly showed a 30% absolute increase in the FFAF from 36% to 66% (***p* = 0.026**) among source patients receiving PVI + EGF-guided ablation vs. PVI-only. Furthermore, since Type II and Type III patients with sources ablated had 66% FFAF, their outcomes mirrored and non-significantly differed from the Type I and Type IV patients who had 65% FFAF after PVI-only (*p* = 0.953). These data are shown in Table 3 and Figure 5.

## 4. Discussion

The randomized, controlled *FLOW-AF* clinical trial represented the first attempt to determine if EGF mapping could be employed in a clinical setting to provide prognostic information and guide the targeted ablation of AF sources in difficult-to-treat redo and PeAF patients. After accomplishing these goals, the *AF-FLOW Global Registry* was subsequently designed to observe utilization of EGF mapping across a wider cohort of physicians treating AF all-comers and to validate the findings from *FLOW-AF*. Over these two studies of different AF populations from different clinical centers, EGF mapping successfully categorized patients into distinct phenotypes based on the presence of extra-PV sources and the quantification of substrate health. 

Patients who recurred had a lower mean EGFC in both *FLOW-AF* and the *AF-FLOW Global Registry*. The presence of an abnormal substrate alone (overall EGFC of ≤0.62) also resulted in a 37% and 41% reduction in FFAF in the two trials, respectively. The relationship established between EGFC and patient outcomes in *FLOW-AF*, therefore, remained consistent among the *AF-FLOW Global Registry* patients.

These differences in FFAF rates based on EGFC can likely be attributed to the importance of tissue homogeneity in propagating a stable electrical wavefront. Substrate complexion can be altered by a variety of different mechanisms, such as abnormal myocyte hypertrophy, fibroblast proliferation, or even rapid inflammation. When such factors are not present, maps have higher EGFC whereby both the overall magnitude of flow vectors are higher and their directionality is more consistent over time, so that when segments are summed over the timescale of a 60 s recording, the net magnitude of flow in each location is minimally cancelled out by changes in direction in subsequent frames. Such patterns of rapid, consistent flow through the substrate suggest that a patient’s AF is more organized and less chaotic, which we find to be a good prognostic sign. To support these findings, correlations have already been found between EGFC and bipolar voltage [31,35], which is also generally linked to improved outcomes; however, it remains a future goal to determine if associations exist between EGFC and localized regions of fibrosis or cardiomyopathy via cardiac MRI. It is also of interest whether SGLT2 inhibitors improve EGFC, as they have been shown to potentially reduce atrial fibrosis and remodeling, as well as decrease AF recurrence [37,38]. Nevertheless, through EGF mapping, physicians can already quantify the structural homogeneity of each patient via their overall EGFC, giving a prognostic assessment of the patient’s level of AF organization or atrial health.

When further analyzing EGFC by atrium across both trials, it followed intuitively that patients had the best prognosis when they had EGFC values above the population mean in both their left and right atria. Accordingly, patients suffered a decreased FFAF when their EGFC was lower in either atrium, and the FFAF rates were the lowest when both the right and left atria concurrently had an EGFC below the population mean. The data also showed that categorizing patients based on the location of low EGFC zones was not as useful as collectively classifying them as either low or high EGFC based on an overall threshold, as previously described in *FLOW-AF* [32]. When overall EGFC is employed as a prognostic metric in this manner, the percentage of healthy substrate over the entire endocardial surface is quantified irrespective of anatomical location. As such, all tissue appears equally important in contributing to the conduction pathways propagating AF and the consequent likelihood of recurrence.

The pooled data comparing LA and RA EGFCs also demonstrated that the LA EGFC was significantly lower than the RA EGFC across all patients; however, there was a correlation between the LA EGFC and RA EGFC, which suggests that the substrate health of one atrium is not independent from the other. The slope of 0.51 further indicates that the already higher EGFC in the RA decreased from patient-to-patient at half the rate as it did in the LA. As such, these observations may collectively reflect that, pathophysiologically, disease originates in the LA near the PVs and spreads to the RA [39]. It thus follows that biatrial disease with maximal substrate remodeling is associated with the worst outcomes.

The ability to prognosticate patient outcomes increased when EGFC quantification was paired with noting the presence or absence of extra-PV sources. These sources were shown to decrease the rates of FFAF in both *FLOW-AF* and the *AF-FLOW Global Registry* after PVI-only therapy. Accordingly, through combining source presence and overall EGFC, four phenotypes were outlined in *FLOW-AF*, and they were again employed in the *AF-FLOW Global Registry* for validation. 

Though small adjustments to the software, phenotype cutoffs, and mapping and ablation strategies have emerged over time, careful efforts were made to align the patients between the two trials when comparing them for the analysis presented here. When first using EGF mapping as a strictly diagnostic tool, we studied patients who did not receive ablation of EGF-identified sources to predict PVI-only outcomes. In both trials, Type I patients had a 100% FFAF at 12 months. Since Type I patients have a healthy substrate and no extra-PV sources, they are predominantly patients who can be treated solely with PVI with reliably good outcomes. When then looking at Type II and Type IV patients, the presence of either extra-PV sources or a low EGFC independently dropped the FFAF rate by a similar magnitude. This trend was maintained across the two trials. Lastly, when comparing the FFAF in Type II and Type IV patients with Type III patients who have both prognostic factors, a multiplicative drop in FFAF was noted due to the presence of both factors. Type III patients were shown to have the lowest rates of FFAF across both studies, likely due to the compounding effects of having a low EGFC and extra-PV sources after PVI-only.

Beyond using EGF as a predictive tool for the prognosis of PVI-only patients, both trials also had a separate treatment population defined as patients who received EGF-guided ablation of extra-PV sources. In *FLOW-AF*, there was a strict cutoff requiring multiple maps to show an SAC of ≥ 26.5% for ablation in the treatment-arm patients, while the mapping and SAC thresholds for extra-PV source ablation varied from patient to patient in the *AF-FLOW Global Registry* based on physician discretion. Still, patients could be aligned to define treated Type II and Type III patients as those with at least one source with SAC ≥ 26.5% ablated, and this EGF-guided ablation of sources increased the FFAF rates in both trials to produce an absolute increase of 30% FFAF. Type II and Type III patients receiving source ablation also had similar and non-significantly different outcomes than those of Types I and Type IV patients who never had sources identified in the first place. Therefore, the ability for targeted ablation to negate the deleterious effects of extra-PV sources indicates that EGF-detected sources may play a significant role in propagating the AF circuits beyond the PVs and suggests that their ablation is necessary for optimal treatment. When combined with improved ablation strategies [9,11], the targeting of all such AF triggers in this manner should continue to improve the effectiveness of therapeutic strategies while providing greater insights into each patient’s mechanism of disease.

A weakness in these comparisons between *FLOW-AF* and the *AF-FLOW Global Registry* is that the patients in the *AF-FLOW Global Registry* were healthier because they included de novo and PAF patients. The sample sizes for PVI-only patients in the Registry were also very small, and even included zero PVI-only Type II patients; however, similar trends in FFAF by phenotype were still present between the two studies, despite the high propensity for population differences or noise. EGF mapping itself also brought limitations, including the need for good wall contact to best assess the health of the substrate and the acute locations of sources. Physicians using the technology additionally needed to be able to steer the basket into a series of positions that covered the full endocardial surface. Still, a diverse group of physicians distinct from those in *FLOW-AF* used EGF mapping in the *AF-FLOW Global Registry* and were able to quickly learn these techniques. Consequently, it seems likely that, with training and skilled use, EGF mapping can provide physicians with an assessment of a patient’s overall substrate health and their presence or absence of extra-PV sources. This information can in turn inform users on a patient’s AF mechanism and how effective a PVI-only or PVI + source ablation approach may be for that patient.

By contrast, current guidelines for classifying AF are based on temporal persistence, where PAF indicates that the longest AF episode lasted fewer than 7 days and PeAF signifies that the longest episode lasted more than 7 days. This framework has been used for outlining inclusion and exclusion criteria for clinical trials and for selecting therapies, but it does not adequately reflect either the actual temporal persistence of AF nor any inherent pathophysiological characteristics, and it is not tied to AF burden or mechanism [40,41,42,43]. We consequently believe that EGF phenotyping provides a superseding means of binning patients based on their pathophysiology that will be more useful in directing treatments. We also expect future efforts to work on refining phenotype cutoffs and the therapeutic strategies employed within each category, particularly in regard to developing an ablation strategy for patients with a low EGFC (Type III and Type IV) that targets the substrate to improve outcomes beyond PVI + source ablation [38,44]. At the same time, tuning an ablation strategy to a patient’s individual phenotype will likely allow for fewer unnecessary ablations and for targeting the minimally necessary treatment to an individual’s specific mechanism of AF.

## 5. Conclusions

The *AF-FLOW Global Registry* was a multicenter trial that validated the results of the randomized, controlled *FLOW-AF* trial. In analyzing both trials, trends were established that demonstrated the efficacy of EGF mapping in providing prognostic information and guiding treatment for AF. Evidence for the following was found: (1) EGFC can be an effective marker of substrate health, and high EGFC values are tied to a higher 12-month FFAF; (2) LA EGFC is lower than but correlated with RA EGFC, suggestive of general substrate remodeling progressive from the LA to the RA; (3) EGF-identified extra-PV sources may be potent drivers of AF, whose presence decreases FFAF rates if left unablated; and (4) phenotyping patients based on EGFC and source presence may be useful in predicting patient recurrence and designing optimal mechanism-specific ablation strategies.

## Figures and Tables

**Figure 1 jcm-14-00693-f001:**
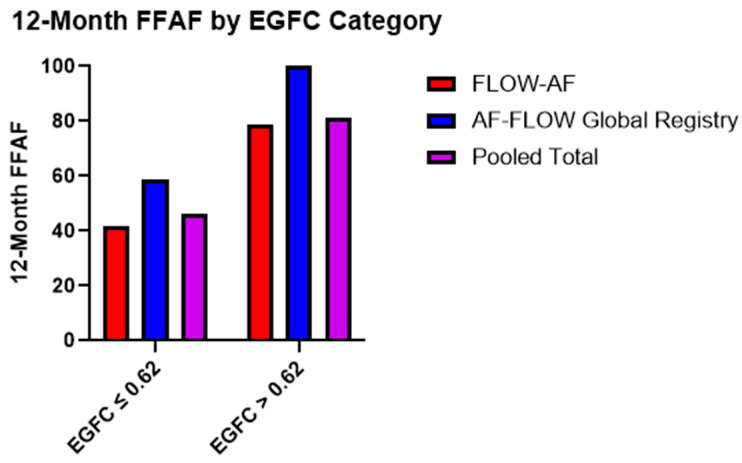
Twelve-month FFAF rates based on EGFC categorization. Patients with healthier substrate (EGFC > 0.62) demonstrated better outcomes (*p* = 0.005) in a consistent pattern across both *FLOW-AF* and the *AF-FLOW Global Registry*.

**Figure 2 jcm-14-00693-f002:**
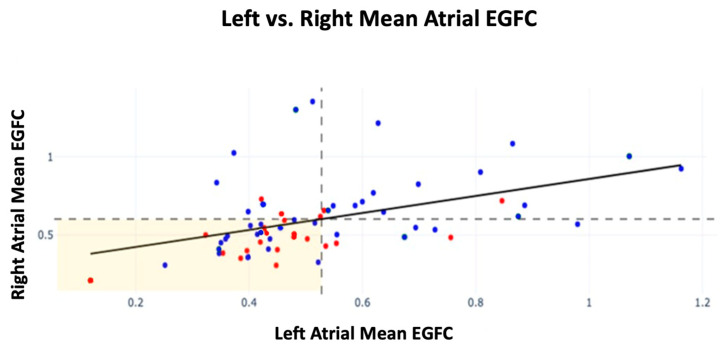
Mean EGFC and outcomes by atrium across all patients in *FLOW-AF* and the *AF-FLOW Global Registry*. FFAF patients are depicted in blue; recurrence patients are depicted in red. The horizontal dashed line indicates the population mean RA EGFC; the vertical dashed line indicates the population mean LA mean EGFC. A linear relationship was established between RA and LA EGFC means (m = 0.51, r = 0.42, *p* < 0.001). Slope and r were unaffected by recurrence status of patients.

**Figure 3 jcm-14-00693-f003:**
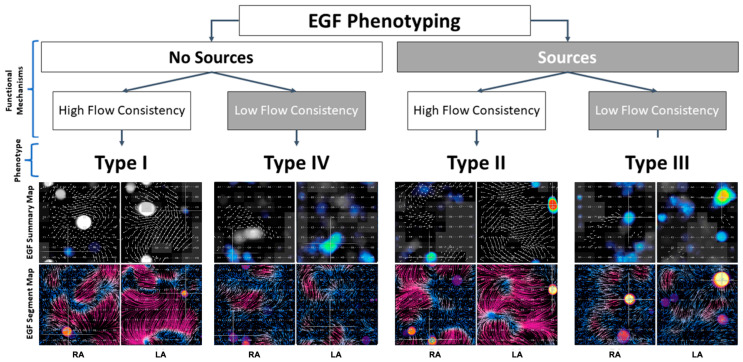
Four phenotypes were characterized based on the presence or absence of EGF-identified sources coupled with the presence of high or low EGFC. Active sources appear as yellow-red spots on summary maps and often localize to focal regions of divergent flow (yellow) on segment maps. EGFC is visualized as high-magnitude flow in a consistent direction over time, so regions with high EGFC are seen as vectors of longer length on summary maps and as purple regions on segment maps. Shorter vectors and more blue regions are areas of disorganized flow with low EGFC.

**Figure 4 jcm-14-00693-f004:**
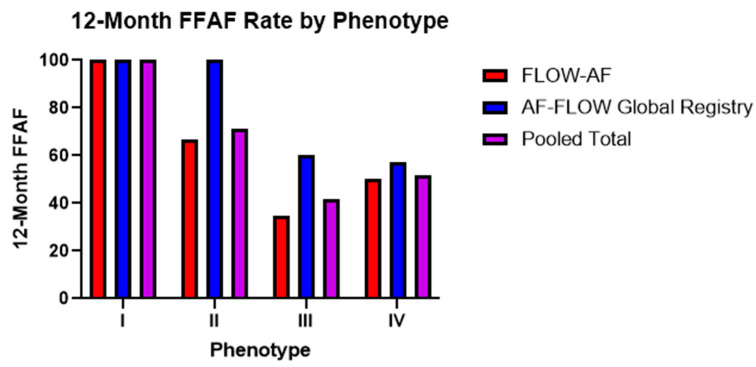
Twelve-month FFAF rates by phenotype across both trials. Trends from the *AF-FLOW Global Registry* help to validate those initially reported in *FLOW-AF* with improved Type II and Type III outcomes in the *AF-FLOW Global Registry* buoyed by the higher rate of EGF-guided source ablation among *AF-FLOW Global Registry* patients.

**Figure 5 jcm-14-00693-f005:**
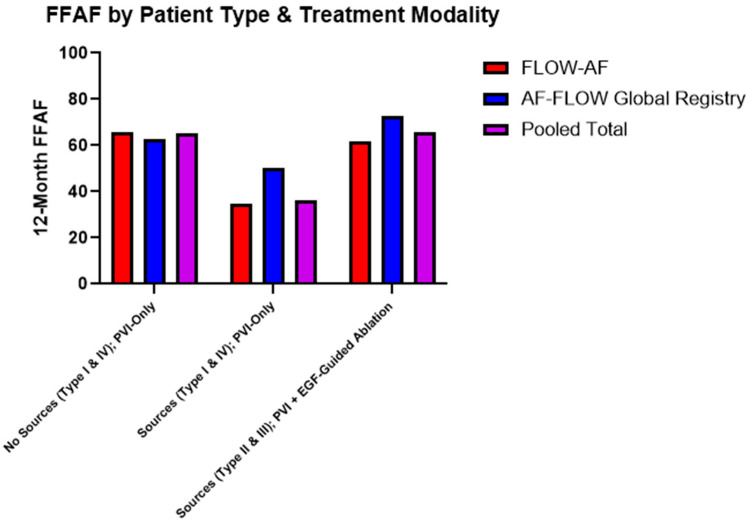
Twelve-month FFAF by patient type and treatment modality. Patients who received extra-PV ablation of EGF-identified sources showed higher rates of FFAF across both trials (*p* = 0.026), bringing FFAF to similar levels observed in patients without sources. Trends in improved outcomes due to source ablation in the *AF-FLOW Global Registry* align with those of *FLOW-AF*.

**Table 1 jcm-14-00693-t001:** Patient demographics in *FLOW-AF* and the *AF-FLOW Global Registry*. There were no significant differences observed between the patient populations of the two trials.

Patient Demographics	*FLOW-AF*	*AF-FLOW Global Registry*	*p*-Value
Age (years)	65.6 ± 9.3	63.9 ± 11.2	0.445
Female sex, n (%)	31/85 (36%)	9/25 (36%)	0.968
LA diameter (cm)	4.4 ± 0.6	4.4 ± 0.5	1.000
Left ventricular ejection fraction (%)	57.2 ± 6.4	55.0 ± 2.5	0.096
Body mass index (kg/m^2^)	29.1 ± 4.6	27.9 ± 3.9	0.239
CHA_2_DS_2_VASc score, mean ± SD	2.4 ± 1.3	2.7 ± 1.9	0.367
Diabetes, n (%)	15/85 (18%)	4/25 (16%)	0.849
Hypertension, n (%)	63/85 (74%)	16/25 (64%)	0.322
History of AFL, n (%)	15/85 (18%)	6/23 (26%)	0.363

**Table 2 jcm-14-00693-t002:** EGFC for patients experiencing 12-month FFAF vs. AF recurrence in *FLOW-AF* and the *AF-FLOW Global Registry*. The FFAF EGFC was significantly higher than the recurrence EGFC (*p* = 0.022).

Trial	FFAF Patient EGFC	Recurrence Patient EGFC
*FLOW-AF*	0.52 ± 0.11 (n = 42)	0.49 ± 0.09 (n = 34)
*AF-FLOW Global Registry*	0.59 ± 0.15 (n = 14)	0.47 ± 0.12 (n = 7)
Pooled total	0.54 ± 0.12 (n = 56)	0.49 ± 0.09 (n = 41)

**Table 3 jcm-14-00693-t003:** FFAF by phenotype and treatment modality for patients enrolled in *FLOW-AF* and the *AF-FLOW Global Registry*. Among PVI-only patients, FFAF decreased significantly between Type I patients with neither sources nor low EGFC to Type II and Type IV patients with either sources or low EGFC (*p* = 0.003) and decreased further between Type I patients and Type III patients with both sources and low EGFC (*p* < 0.001).

Patient Phenotype and Treatment	*FLOW-AF* FFAF	*AF-FLOW Global Registry* FFAF
Type I—PVI-only	10/10 (100%)	1/1 (100%)
Type II—PVI-only	5/10 (50%)	n/a
Type III—PVI-only	3/13 (23%)	1/2 (50%)
Type IV—PVI-only	11/22 (50%)	4/7 (57%)
Type II—PVI + EGF-guided ablation	7/8 (88%)	3/3 (100%)
Type III—PVI + EGF-guided ablation	6/13 (46%)	5/8 (63%)

## Data Availability

Data from both *FLOW-AF* (NCT04473963) and the *AF-FLOW Global Registry* (NCT05481359) are available on clinicaltrials.gov.

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
