# Peer review of "Electrographic Flow Mapping Provides Prognosis for AF Ablation Outcomes Across Two Independent Prospective Patient Cohorts"

_jcm, 2025, doi:10.3390/jcm14030693_

Round 1
Reviewer 1 Report
Comments and Suggestions for Authors
The paper is interesting and I would like to congratulate authors for the very good paper They found that patients with high Electrographic Flow Consistency and no sources post-procedure had the best outcomes. EGF mapping provides insights into underlying disease pathophysiology and may be employed prospectively to predict recurrence. I have only minor comments to improve the manuscript:
Using CARTO and Navex, authors should more discuss other fundamental tools that may also impact on recurrence, focusing on combination of highly localized impedance (LI) and contact force that may improve tissue characterization and lesion prediction during radiofrequency (RF) pulmonary vein isolation (DOI: 10.3389/fcvm.2023.1169037). Please discuss and cite suggested reference
In discussion authors should also cite importance of novel energies, and its possible interactions with this patient population, more discussing about Pulsed field ablation (PFA), as source that has been shown to be tissue selective and is expected to decrease damage to non-cardiac tissue while providing high efficacy in pulmonary vein isolation (DOI: 10.15420/aer.2022.45) Please discuss and cite suggested reference
Did authors considered patients with concomitant cardiomyopathies? Plase discuss more about this important point and its potential relationship with ECG mapping
Author Response
Comments 1: The paper is interesting and I would like to congratulate authors for the very good paper They found that patients with high Electrographic Flow Consistency and no sources post-procedure had the best outcomes. EGF mapping provides insights into underlying disease pathophysiology and may be employed prospectively to predict recurrence. I have only minor comments to improve the manuscript:
Using CARTO and Navex, authors should more discuss other fundamental tools that may also impact on recurrence, focusing on combination of highly localized impedance (LI) and contact force that may improve tissue characterization and lesion prediction during radiofrequency (RF) pulmonary vein isolation (DOI: 10.3389/fcvm.2023.1169037). Please discuss and cite suggested reference
Response 1: Please note revisions for this reviewer are highlighted in yellow in the manuscript. We thank the reviewer for his or her careful reading of the paper. We agree that the paper lacked discussion of methods to improve PVI and instead had jumped quickly into discussing ablation targets beyond PVI. We have therefore rectified this gap by adding discussion of the suggested paper on lines 57-63 of the Introduction and an additional reference to it on line 429 of the Discussion. We also will note here that the need for good contact during ablations matches our findings of a need for good wall contact during EGF mapping and ablation, as noted in Section 2.5 of the Methods and added to the Discussion on line 438.
Comments 2: In discussion authors should also cite importance of novel energies, and its possible interactions with this patient population, more discussing about Pulsed field ablation (PFA), as source that has been shown to be tissue selective and is expected to decrease damage to non-cardiac tissue while providing high efficacy in pulmonary vein isolation (DOI: 10.15420/aer.2022.45) Please discuss and cite suggested reference
Response 2: In the same vein, a discussion of PFA was also lacking from the Introduction, which was added on lines 63-68 and guided by the suggested review article. The reference was also cited on line 429 of the Discussion. We will keep aware of recent stroke and hemolysis data with PFA before publication of the final version.
Comments 3: Did authors considered patients with concomitant cardiomyopathies? Plase discuss more about this important point and its potential relationship with ECG mapping
Response 3: We thank the reviewer for this interesting suggestion. Unfortunately, however, we did not explicitly ask or screen for cardiomyopathy in our trials and have limited accounts of cardiomyopathy in the provided medical history. Those patients that do have cardiomyopathy were not significantly associated with source presence or absence or with high or low EGFC, though this remains an area of interest for future research, especially comparing EGF maps to cardiac MRIs. A mention of this goal has been added on lines 366-369 of the Discussion.
Reviewer 2 Report
Comments and Suggestions for Authors
This manuscript explores the utility of Electrographic Flow (EGF) mapping and Electrographic Flow Consistency (EGFC) in predicting atrial fibrillation (AF) ablation outcomes. It compares findings from the FLOW-AF trial with those of the AF-FLOW Global Registry, focusing on the prognostic value of EGF mapping in categorizing patients by substrate health and source presence. While the study offers valuable insights into the prognostic capabilities of EGF mapping, it requires further elaboration on certain methodological aspects, clarification of results, and a broader discussion to enhance its clinical relevance.
The introduction adequately emphasizes the challenges of managing AF, particularly persistent AF (PeAF), and highlights the potential role of EGF mapping. However, it would benefit from:
· To improve the rationale, the authors should expand on the adverse prognosis of the AF population as highlighted in this very recent paper published in EHJ 10.1093/eurheartj/ehae694.
· Providing a succinct overview of current limitations in mapping technologies.
· Clearly stating the research objectives and hypotheses.
The methods section outlines the use of EGF mapping and EGFC calculation but lacks crucial details:
· Include a more comprehensive explanation of the algorithm used for EGF mapping and its validation.
· Specify criteria for defining phenotypes, particularly threshold values used to differentiate between high and low EGFC.
· Clarify the rationale behind the sample size and the inclusion criteria for each cohort.
The results effectively compare patient outcomes across the FLOW-AF and AF-FLOW Global Registry studies. However:
· Present key data in a consolidated table summarizing outcomes by phenotype across both studies.
· Highlight statistically significant differences more explicitly.
· Include a more detailed analysis of variability in results due to differences in study populations and methodologies.
Discussion, the following are required:
· A more robust exploration of potential mechanisms by which EGFC predicts outcomes.
· Expanded discussion on the clinical implications of phenotyping patients using EGF mapping.
· Consideration of the role of emerging therapeutic agents, such as SGLT2 inhibitors, which have shown promise in AF management.
· Analysis of the limitations of EGF mapping, including operator variability and technology-related constraints.
Review for grammatical consistency and adherence to journal style guidelines.
Author Response
Comments 1: This manuscript explores the utility of Electrographic Flow (EGF) mapping and Electrographic Flow Consistency (EGFC) in predicting atrial fibrillation (AF) ablation outcomes. It compares findings from the FLOW-AF trial with those of the AF-FLOW Global Registry, focusing on the prognostic value of EGF mapping in categorizing patients by substrate health and source presence. While the study offers valuable insights into the prognostic capabilities of EGF mapping, it requires further elaboration on certain methodological aspects, clarification of results, and a broader discussion to enhance its clinical relevance.
The introduction adequately emphasizes the challenges of managing AF, particularly persistent AF (PeAF), and highlights the potential role of EGF mapping. However, it would benefit from:
- To improve the rationale, the authors should expand on the adverse prognosis of the AF population as highlighted in this very recent paper published in EHJ 10.1093/eurheartj/ehae694.
- Providing a succinct overview of current limitations in mapping technologies.
- Clearly stating the research objectives and hypotheses.
Response 1: Please note revisions for this reviewer are highlighted in green in the manuscript. We thank the reviewer for his or her careful reading of the paper. We agree that the Introduction had lacked these elements and appreciate the reviewer’s suggestions to rectify them. We have added a discussion surrounding the disease burden of AF focused on the suggested paper on lines 42-49 of the Introduction. A listing of the limitations of mapping technologies has been added to lines 76-81 which segues into the prior discussion on how how EGF mapping differs. Lastly, objective have been added on lines 104-107.
Comments 2: The methods section outlines the use of EGF mapping and EGFC calculation but lacks crucial details:
- Include a more comprehensive explanation of the algorithm used for EGF mapping and its validation.
- Specify criteria for defining phenotypes, particularly threshold values used to differentiate between high and low EGFC.
- Clarify the rationale behind the sample size and the inclusion criteria for each cohort.
Response 2: We thank the reviewer for his or her suggestions and agree that these inclusions would improve the Methods of the paper. EGFC thresholds have been further detailed on lines 179-181. Notes on trial sample sizes and powering were also added on lines 113-116. Importantly, neither trial was initially individually powered to show outcome differences by trial arm. For the EGF algorithm details, we believe this question is quite expansive and is best left referred to in other publications. We have accordingly expanded upon lines 138-141 to now include a citation to both the technical and mathematical foundations behind EGF mapping as well as a recent review article with a three-paragraph summary. Still, if the reviewer and/or editor believe any of these details should be included in the present manuscript, we would be happy to add those that are most critical.
Comments 3: The results effectively compare patient outcomes across the FLOW-AF and AF-FLOW Global Registry studies. However:
- Present key data in a consolidated table summarizing outcomes by phenotype across both studies.
- Highlight statistically significant differences more explicitly.
- Include a more detailed analysis of variability in results due to differences in study populations and methodologies.
Response 3: We thank the reviewer for these suggestions to improve the clarity of the Results, as well. An outcomes table (Table 3) has been added on lines 280-284. We have also bolded results where p<0.05, which we think improves readability of these results if acceptable to the editors. We additionally added comments on significant findings to table and figure legends. Regarding result variability, the Results were retuned to address this, primarily on lines 302-313. We have also taken efforts to bin patients in Table 3 and the plots that follow based on the protocol that most closely aligns with the equivalent protocol in FLOW-AF so that after this binning, only minor steps, e.g. triplicate pre-ablation mapping, may not have aligned between the two studies. With regards to the patient population, we would like to emphasize that a goal of this manuscript is to show that EGF phenotyping works for allcomers regardless of conventional AF type, which is a crude assessment determined by length of documented AF episodes. We therefore think a PeAF-only sub-analysis of the AF-FLOW Global Registry would take away from this message and would be best left contained in the manuscript breaking down the AF-FLOW Global Registry results alone. A discussion of all these points has been added to the Discussion on lines 401-403, 418-422, and 436-463.
Comments 4: Discussion, the following are required:
- A more robust exploration of potential mechanisms by which EGFC predicts outcomes.
- Expanded discussion on the clinical implications of phenotyping patients using EGF mapping.
- Consideration of the role of emerging therapeutic agents, such as SGLT2 inhibitors, which have shown promise in AF management.
- Analysis of the limitations of EGF mapping, including operator variability and technology-related constraints.
Response 4: We thank the reviewer for his or her suggestions to expand on these Discussion topics. We have expanded upon and highlighted the purported mechanism by which EGFC predicts outcomes, discussing the vector implications in the EGF maps and drawing on correlations to bipolar voltage on lines on lines 360-387. SGLT2 inhibitors were mentioned in this discussion on lines 369-371, and they were additionally cited as a potential therapy for low EGFC along with low voltage zone ablation on line 460. For phenotyping with EGF mapping, we concluded our discussion with this needed exploration on lines 449-463. Limitations of mapping including operator skill were also emphasized on lines 436-448.
Comments 5: Review for grammatical consistency and adhere to journal’s style guidelines
Response 5: We thank the reviewer for the advice. General editorial changes have been highlighted in blue. We have also scanned the paper for consistent use of hyphens, Oxford commas, passive voice, etc., and we have made changes when appropriate.
Round 2
Reviewer 1 Report
Comments and Suggestions for Authors
manuscript definitely improved. Congratulations to authors for this very good paper
Reviewer 2 Report
Comments and Suggestions for Authors
The authors have appropriately addressed the raised concerns